# LACMA: Language-Aligning Contrastive Learning with Meta-Actions for Embodied Instruction Following

**Cheng-Fu Yang**[1*], **Yen-Chun Chen**[2], **Jianwei Yang**[2], **Xiyang Dai**[2],
**Lu Yuan**[2], **Yu-Chiang Frank Wang**[3,4], **Kai-Wei Chang**[1]

[1]UCLA     [2]Microsoft     [3]National Taiwan University     [4]Nvidia

{cfyang, kwchang}@cs.ucla.edu     frankwang@nvidia.com

{yen-chun.chen, jianwei.yang, xiyang.dai, luyuan}@microsoft.com

## Abstract

End-to-end Transformers have demonstrated an impressive success rate for Embodied Instruction Following when the environment has been seen in training. However, they tend to struggle when deployed in an unseen environment. This lack of generalizability is due to the agent's insensitivity to subtle changes in natural language instructions. To mitigate this issue, we propose explicitly aligning the agent's hidden states with the instructions via contrastive learning. Nevertheless, the semantic gap between high-level language instructions and the agent's low-level action space remains an obstacle. Therefore, we further introduce a novel concept of *meta-actions* to bridge the gap. Meta-actions are ubiquitous action patterns that can be parsed from the original action sequence. These patterns represent higher-level semantics that are intuitively aligned closer to the instructions. When meta-actions are applied as additional training signals, the agent generalizes better to unseen environments. Compared to a strong multi-modal Transformer baseline, we achieve a significant **4.5**% absolute gain in success rate in unseen environments of ALFRED Embodied Instruction Following. Additional analysis shows that the contrastive objective and meta-actions are complementary in achieving the best results, and the resulting agent better aligns its states with corresponding instructions, making it more suitable for real-world embodied agents.[1]

## 1 Introduction

Embodied Instruction Following (EIF) necessitates an embodied AI agent to interpret and follow natural language instructions, executing multiple sub-tasks to achieve a final goal. Agents are instructed to sequentially navigate to locations while localizing and interacting with objects in a fine-grained

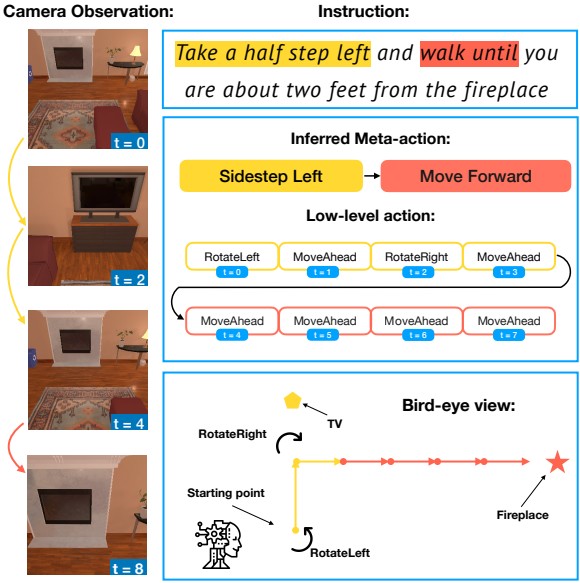

Figure 1: An embodied agent takes camera observations and instructions and then execute actions to fulfill a goal. The large semantic gap between the instruction "Take a half step left" and the action sequence "RotateLeft, MoveAhead, RotateRight, MoveAhead" may cause the agent to learn shortcuts to the camera observation, ignoring the language instruction. We propose semantic meaningful meta-actions to bridge this gap.

manner. In a typical EIF simulator, the agent's sole perception of the environment is through its egocentric view from a visual camera. To complete all the sub-tasks in this challenging setting, variants of Transformer (Vaswani et al., 2017) have been employed to collectively encode the long sequence of multi-modal inputs, which include language instructions, camera observations, and past actions. Subsequently, the models are trained end-to-end to imitate the ground-truth action sequences, *i.e.*, expert trajectories, from the dataset.[2]

Significant progress has been made in this field (Pashevich et al., 2021; Suglia et al., 2021; Zhang and Chai, 2021). Nevertheless, our obser-

---

*work partially done as a research intern at Microsoft

[1]The code is available at: github.com/joeyy5588/LACMA.

[2]We use action sequences and trajectories interchangeably.

vations suggest that existing approaches might not learn to follow instructions effectively. Specifically, our analysis shows that existing models can achieve a high success rate even *without* providing any language instructions when the environment in the test time is the same as in training. However, performance drops significantly when they are deployed into an unseen environment even when instructions are provided. This implies that the models learn to memorize visual observations for predicting action sequences rather than learning to follow the instructions. We hypothesize that this overfitting of the visual observations is the root cause of the significant performance drop in *unseen* environments. Motivated by this observation, we raise a research question: *Can we build an EIF agent that reliably follows instructions step-by-step?*

To address the above question, we aim to improve the alignment between the language instruction and the internal state representation of an EIF agent. Sharma et al. (2021) suggest leveraging language as intermediate representations of trajectories. Jiang et al. (2022) demonstrate that identifying patterns within trajectories aids models in adapting to unseen environments. Inspired by their observations, we conjecture two critical directions: 1) language may be used as a pivot, and 2) common language patterns across trajectories could be leveraged. We propose **L**anguage-**A**ligning **C**ontrastive Learning with **M**eta-**A**ctions (LACMA), a method aimed at enhancing Embodied Instruction Following. Specifically, we explicitly align the agent's hidden states, which are employed in predicting the next action, with the corresponding sub-task instruction via contrastive learning. Through the proposed contrastive training, hidden states are more effectively aligned with the language instruction.

Nevertheless, a significant semantic gap persists between high-level language instructions, *e.g.*, "take a step left and then walk to the fireplace" and the agent's low-level action space, *e.g.*, MoveForward, RotateRight, *etc*. To further narrow this gap, we introduce the concept of *meta-actions* (MA), a set of action patterns each representing higher-level semantics, and can be sequentially composed to execute any sub-task. For clarity, we will henceforth refer to the original agent's actions as low-level actions (LA) for the remainder of the paper. The elevated semantics of meta-actions may serve as a more robust learning signal, preventing the model from resorting to shortcuts based on

its visual observations. This concept draws inspiration from recent studies that improve EIF agents with human-defined reusable skills (Brohan et al., 2022; Ahn et al., 2022). An illustrative example is shown in Figure 1.

More specifically, our agent first predicts meta-actions, and then predicts low-level actions conditioning on the MAs. However, an LA sequence may be parsed into multiple valid MA sequences. To determine the optimal MA sequences, we parse the trajectories following the minimum description length principle (MDL; Grünwald, 2007). The MDL states that the shortest description of the data yields the best model. In our case, the optimal parse of a trajectory corresponds to the shortest MA sequence in length. Instead of an exhaustive search, optimal MAs can be generated via dynamic programming.

To evaluate the effectiveness of LACMA, we conduct experiments on the ALFRED dataset (Shridhar et al., 2020). Even with a modest set of meta-actions consisting of merely 10 classes, our agent significantly outperforms in navigating unseen environments, improving the task success rate by $4.7\%$ and $4.5\%$ on the unseen validation and testing environments, respectively, while remaining competitive in the seen environments. Additional analysis reveals the complementary nature of the contrastive objective and meta-actions: Learning from meta-actions effectively reduces the semantic gap between low-leval actions and language instructions, while the contrastive objective enforces alignment to the instructions, preventing the memorization of meta-action sequences in seen environments.

Our contributions can be summarized as follows:

- We propose a contrastive objective to align the agent's state representations with the task's natural language instructions.

- We introduce the concept of meta-actions to bridge the semantic gap between natural language instructions and low-level actions. We also present a dynamic programming algorithm to efficiently parse trajectories into meta-actions.

- By integrating the proposed language-aligned meta-actions and state representations, we enhance the EIF agents' ability to faithfully follow instructions.

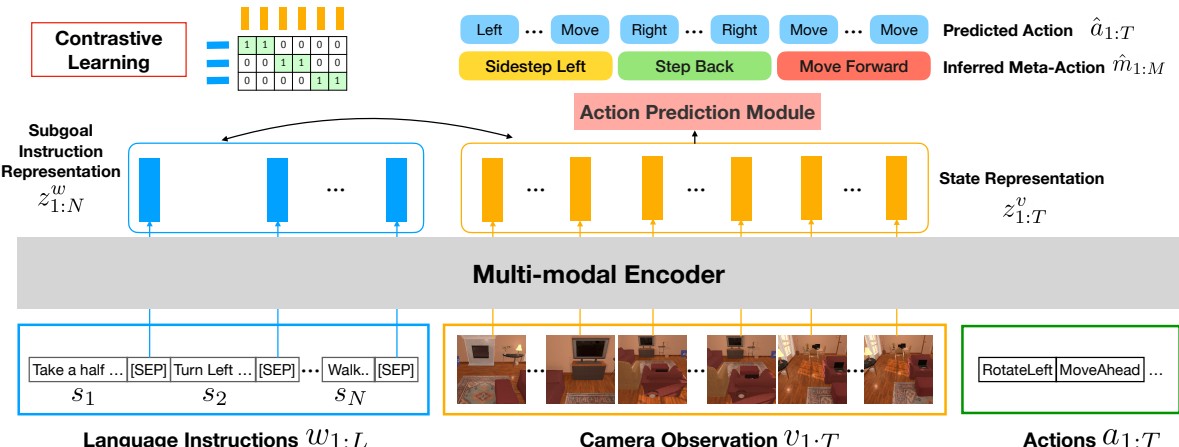

Figure 2: LACMA takes instructions $w_{1:L}$, camera observations $v_{1:T}$, and actions $a_{1:T}$ as inputs, and then output state representations $z_{1:T}^v$. We additionally extract the output features corresponding to the [SEP] tokens as the representations for the sub-goal instructions $z_{1:N}^w$. $z_{1:T}^v$ and $z_{1:N}^w$ are used for contrastive learning, while $z_{1:T}^v$ are further utilized to predict the meta-action sequences $\hat{m}_{1:M}$ and the low-level action sequences $\hat{a}_{1:T}$.

## 2 Method

In this section, we first define settings and notations of the embodied instruction following (EIF) tasks in Section 2.1. Then, in Section 2.2, we introduce the language-induced contrastive objective used to extract commonalities from instructions. Finally, in Section 2.3, we will explain how we generate the labels for meta-actions and how they are leveraged to bridge the gap between instructions and the corresponding action sequences.

### 2.1 Settings and Notations

Given a natural language task goal $G$ which consists of $N$ sub-goals, each corresponding to a sub-goal instruction $S = s_{1:N}$. The agent is trained to predict a sequence of executable low-level actions $a_{1:T}$ to accomplish the task. During training time, the ground-truth trajectories of the task are represented by the tuple $(w_{1:L}, v_{1:T}, a_{1:T})$, where $T$ denotes the length of the trajectories. $w_{1:L}$ represents the concatenation of the task description $G$ and all the subgoal instructions $S_{1:N}$, with each instruction appended by a special token [SEP]. $L$ stands for the total number of tokens in the concatenated sentence. $v_{1:T}$ denotes the camera observations of the agent over $T$ steps, with each camera frame $v_t$ being an RGB image with a spatial size of $W \times H$, denoted as $v_t \in \mathbb{R}^{W \times H \times 3}$. The action sequences $a_{1:T}$ denote the ground-truth actions. At each timestep $t$, the navigation agent, parameterized by $\theta$, is trained to optimize the output distribution $P_\theta(a_t | w_{1:L}, v_{1:t}, a_{t-1})$. An overview of the framework is illustrated in Figure 2.

### 2.2 Contrastive State-Instruction Alignment

In Section 1, we put forth the hypothesis that tasks with similar objectives or navigation goals exhibit shared language patterns. Extracting such commonalities can effectively facilitate the alignment between language instructions and action sequences, further enhancing the generalizability of acquired skills. This alignment is further reinforced through the utilization of a contrastive objective during training. In this subsection, we first describe the process of obtaining the model's state representation, which encapsulates the relevant information necessary for the agent to make decisions and take actions. We then explain how we associate such representations with linguistic features to construct both positive and negative pairs for contrastive learning.

**State and Instruction Representations** Following prior studies (Pashevich et al., 2021; Suglia et al., 2021; Zhang and Chai, 2021), we use a Transformer encoder to process all input information, which includes the input instructions, camera observations and previously executed actions $(w_{1:L}, v_{1:t}, a_{t-1})$. As shown in Figure 3, our model generates the state representation $z_t^v$ at each timestep $t$, which captures the current state of the agent and the environment. To extract representations for each sub-goal, we take the output features of the [SEP] tokens appended after each instruction, resulting in $N$ features $z_{1:N}^w$. Please refer to the appendix for more details.

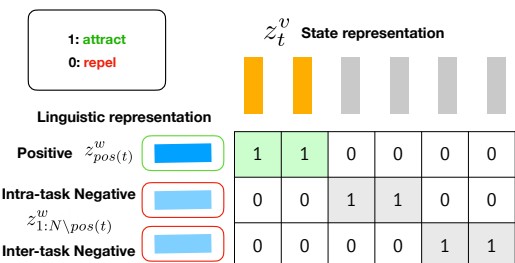

| Meta-Actions | Regular Expressions | Example Action Sequences |
|---|---|---|
| Step Left | "lm{,3}r" | l, m, m, r |
| Move Forward | "m{1,}" | m, m, m |
| Step Back | "(ll\|rr)m+(ll\|rr)" | l, l, m, r, r |

Table 1: Examples of meta-actions. "l", "m", and "r" represent RotateLeft, MoveAhead, and RotateRight, respectively. Due to the page limit, we leave the full meta-action list int Table 8 in the appendix.

Figure 3: We contrasts a single positive $z^w_{pos(t)}$ (the corresponding language instruction) for each state representation $z^v_t$ against a set of *intra-task* negatives (other instructions from the same task $G$) and *inter-task* negatives (instructions from other tasks).

**Constructing Positive and Negative Pairs** As illustrated in Fig. 3, our contrastive loss function compares a specific positive pair, consisting of a state representation $z^v_t$ and the feature of its corresponding subgoal instruction $z^w_{pos(t)}$, with a collection of negative pairs. $pos(t)$ is the index of the instruction features corresponding to state $t$. This mapping ensures that each frame at timestamp is correctly aligned with its corresponding language instructions. The negative pairs include other subgoal instructions from the same task (*intra-task* negatives) as well as instructions from different tasks (*inter-task* negatives). We denote these instructions as $z^w_{1:N \setminus pos(t)}$. The contrastive objective takes the following form:

$$\mathcal{L}_{CL} = -\sum_{t=1}^{T} \log \frac{\exp(\langle z^v_t, z^w_{pos(t)} \rangle / \tau)}{\sum_{n=1}^{N} \exp(\langle z^v_t, z^w_n \rangle / \tau)}, \quad (1)$$

where $\langle \cdot, \cdot \rangle$ denotes the inner product and $\tau$ is the temperature parameter. By contrasting the positive pair with these negative pairs, our contrastive loss $\mathcal{L}_{CL}$ encourages the model to better distinguish and align state representations with the language instructions, which allows our model to transfer the learned knowledge from seen environments to unseen environments more effectively.

### 2.3 Learning with Meta-Actions (MA)

To bridge the semantic gap between natural language instructions and navigation skills, we propose the concept of *meta-action* (MA), representing higher-level combinations of low-level actions (LAs), as depicted in Fig. 1. In this subsection, we first introduce how we determine the optimal meta-action sequence given a low-level action trajectory and a set of pre-defined meta-actions. Next, we detail our training paradigm, which involves both generated MAs and the ground-truth LAs.

**Optimal Meta-Actions** We draw an analogy between the minimum description length principle (MDL; Grünwald, 2007) and finding the optimal MA sequences for a given LA trajectory. Both approaches share a common goal: finding the most concise representation of the data. The MDL principle suggests that the best model is the one with the shortest description of the data. Similarly, we aim to find MA sequences that are compact yet lossless representations of LA trajectories. Therefore, we define the optimal meta-action sequence as the one that uses the minimal number of MAs to represent the given low-level action trajectory.

**MA Identification via Dynamic Programming** We formulate the process of finding the minimal number of MAs to represent a given LA trajectory as a dynamic programming (DP) problem. The high level idea is to iteratively solve the subproblem of the optimal MA sequence up to each LA step. To achieve this, we first convert the LA sequence into a sequence of letters and then string match the regular expression form of the given MA set. Table 1 showcases some of the conversions. For instance, the LA sequence "MoveAhead, MoveAhead, MoveAhead" is written as "m, m, m", and the MA "Move Forward" is represented as "m{1,}" (m appears one or more times consecutively). Next, we sequentially solve the subproblem for each time step, finding the optimal MA sequence to represent the LA trajectory up until the current time step. Further details of the algorithm, pseudo code, pre-defined meta-actions, regular expressions, and example low-level action sequences are provided in the appendix A.3 and A.2. By formulating the meta-action identification as a DP problem, we can efficiently extract the optimal meta-action sequence $m_{1:M}$ with a length of $M$ to represent any low-level action sequence.

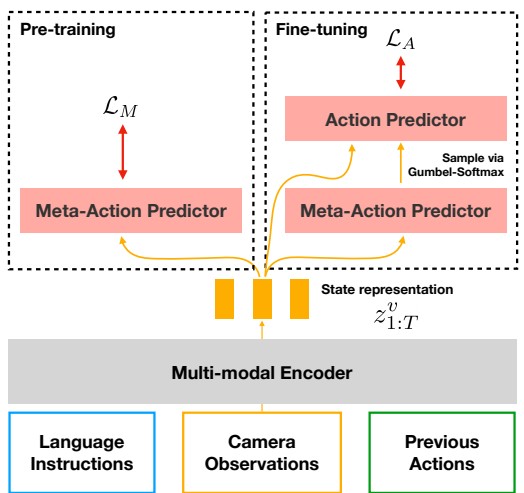

Figure 4: Two-stage training of LACMA. In the pre-training stage, our model is optimized with the DP-labeled meta-actions (MAs). In fine-tuning, we use the ground-truth low-level actions (LAs) as supervision, and the model predicts LAs from its own prediction of MAs. To jointly optimize the MA predictor, we apply Gumbel-softmax to allow gradients to flow through the sampling process of MAs.

**Training Strategies** As shown in Fig. 4, we adopt the pretrain-finetune paradigm to train our model. Specifically, in the initial pre-training stage, we optimize the model using DP-labeled meta-action sequences together with Eqn. (1), the contrastive objective. The objective function of optimizing MA is the standard classification loss: $\mathcal{L}_M = \text{CrossEntropy}(m_{\hat{1}:M}, m_{1:M})$, thus the pre-training loss can be written as $\mathcal{L}_p = \mathcal{L}_{CL} + \mathcal{L}_M$. In the fine-tuning stage, we utilize ground-truth LA sequences as supervision. Similarly, the fine-tuning loss can be written as: $\mathcal{L}_f = \mathcal{L}_{CL} + \mathcal{L}_A$, where $\mathcal{L}_A = \text{CrossEntropy}(a_{\hat{1}:T}, a_{1:T})$ denotes the loss function of LA prediction. The use of $\mathcal{L}_{CL}$ in both stage enforces our model to align the learned navigation skills with the language instructions, preventing it from associating specific visual patterns or objects with certain actions.

However, conditioning LA prediction on DP-labeled MAs might cause exposure bias (Ranzato et al., 2016). This can result in diminished performance in testing, where MAs are predicted rather than being explicitly labeled. To mitigate this train-test mismatch, we employ Gumbel-softmax (Jang et al., 2017), allowing our model to condition on *predicted* MAs for LA prediction during training.

## 3    Related Works

**Embodied Instruction Following (EIF)** Various benchmarks (Anderson et al., 2018b; Pejsa

et al., 2016; Misra et al., 2018; Ku et al., 2020; Krantz et al., 2020; Das et al., 2018; Prabhudesai et al., 2020; Padmakumar et al., 2022; Gao et al., 2022) and environments (Ramakrishnan et al., 2021; Kolve et al., 2017; Li et al., 2023; Savva et al., 2019) have been proposed to study embodied intelligent agents. Among them, vision-and-language navigation (VLN) is the most comparable task to our setting. Various models have demonstrated impressive performance on the task of VLN (Ke et al., 2019; Chen et al., 2021; Jain et al., 2019; Tan et al., 2019; Zhu et al., 2020; Li et al., 2019; Zhu et al., 2021; Schumann and Riezler, 2022). In addition, Liang et al. (2022) contrasted data within the same modality to improve robustness on variations of instructions and visual scenes. We specifically focus on the ALFRED dataset (Shridhar et al., 2020) as it not only involves longer episodes of navigation but also requires models to understand complex instructions, perform fine-grained grounding, and interact with objects.

**Neural EIF Agents** In recent years, two lines of works have been developed to tackle embodied instruction following tasks: modular and end-to-end methods. Modular methods (Min et al., 2022; Blukis et al., 2022; Inoue and Ohashi, 2022) employ multiple modules trained with specific sub-tasks and direct supervision to decompose the EIF tasks. While our work focuses on aligning the state representations with language instructions for improved generalization, modular methods do not produce state representations for task planning. Therefore, we focus specifically on end-to-end methods (Shridhar et al., 2020; Suglia et al., 2021; Zhang and Chai, 2021; Pashevich et al., 2021; Nguyen et al., 2021) to address these limitations. These methods generally utilize a single neural network to directly predict low-level actions from input observations. However, these methods generally suffer from limited interpretability and generalization (Eysenbach et al., 2022). On the other hand, LACMA aligns the decision making process with language instructions, simultaneously enhancing interpretability and generalization.

**Skill Learning** Learning skills from demonstrations has been an active research area in the field of machine learning and robotics. Several approaches have been proposed to acquire skills, including the use of latent variable models to partition the experience into different skills (Kim et al., 2019; Jiang

| Method | Seen | | | | Unseen | | | |
|---|---|---|---|---|---|---|---|---|
| | Val | | Test | | Val | | Test | |
| | SR | GC | SR | GC | SR | GC | SR | GC |
| SEQ2SEQ (Shridhar et al., 2020) | 3.1 | 10.0 | 4.0 | 9.4 | 0.0 | 6.9 | 0.4 | 7.0 |
| MOCA (Singh et al., 2021) | 25.9 | 34.9 | 22.1 | 28.3 | 5.4 | 16.2 | 5.3 | 14.3 |
| EmBERT (Suglia et al., 2021) | **37.4** | **44.6** | 31.8 | 39.2 | 5.7 | 15.9 | 7.5 | 16.3 |
| E.T.[†] (Pashevich et al., 2021) | 34.7 | 42.0 | 28.9 | 36.3 | 3.5 | 13.6 | 4.7 | 14.9 |
| LACMA | 36.9 | 42.8 | **32.4** | **40.5** | **8.2** | **18.0** | **9.2** | **20.1** |

Table 2: Results on ALFRED. SR and GC denote the task success rate and the goal condition success rate, respectively. For path-length-weighted scores, please see Table 10. ([†]: exclude data from unseen environments.)

et al., 2022; Ajay et al., 2021; Tanneberg et al., 2021). Other works focus on learning skills from language supervision (Ahn et al., 2022; Pashevich et al., 2021; Andreas et al., 2018; Sharma et al., 2021; Fried et al., 2018). However, there remains a challenge in bridging the gap between the learned latent skills and natural language. To close the gap, we introduce the concept of meta-actions, which are higher-level actions that captures the semantic meaning of actions in relation to instructions.

## 4 Experiments

### 4.1 Experimental Settings

**Dataset** The ALFRED dataset (Shridhar et al., 2020) comprises demonstrations where an agent completes household tasks based on goals specified in natural language. ALFRED consists of 21,023 train, 1,641 validation (820 seen / 821 unseen), and 3,062 test (1,533 seen / 1,529 unseen) episodes.

**Evaluation Metrics** Following Shridhar et al. (2020), we report the task success rate (SR) and the goal condition success rate (GC). SR measures the percentage of tasks where the agent successfully accomplishes all the subgoals, while GC is the ratio of subgoals fulfilled at the end of the task. For example, the task "put a hot potato slice on the counter" consists of four goal-conditions: slicing the potato, heating the potato slice, placing it on the counter, and ensuring it is both heated and on the counter. A task is considered success only if all the goal-conditions are successful.

**Implementation Details** Our method was built upon Episodic Transformer (E.T.; Pashevich et al., 2021). Specifically, BERT (Devlin et al., 2019) is used to extract features from the language instructions. For visual observations, we pre-train a ResNet-50 Faster R-CNN (Girshick, 2015) on

the ALFRED dataset and then use the ResNet backbone to extract image features. These inputs from different modalities are then fused by a multi-modal Transformer encoder. More training details can be found in the appendix A.1.

**Baselines** As discussed in Sec. 3, LACMA focuses on aligning state representations with language instructions. For fair comparisons, we specifically choose end-to-end methods that do not incorporate an explicit planner, including SEQ2SEQ (Shridhar et al., 2020), MOCA (Singh et al., 2021), EmBERT (Suglia et al., 2021), and Episodic Transformer (E.T.; Pashevich et al., 2021). Note that the original E.T. was trained using additional trajectories from the *unseen* environments, which violates our assumption. Therefore, we reproduce the model using only the data from the original training set.

### 4.2 Quantitative Results

The results on ALFRED are shown in Table 2. We can see that LACMA performed favorably against the best end-to-end models across different metrics. LACMA substantially improves the task success rates (SR) and goal condition success rates (GC), especially in the *unseen* environments. On the validation split, our method outperforms the baseline (E.T.) by $4.7\%$ in SR, and on the test split by $4.5\%$. This verifies our design in aligning the learned skills with language instructions and using meta-actions to bridge the semantic gap between instructions and low-level actions. Moreover, in the seen environments, LACMA not only exhibits improvements compared to the baseline, but is also comparable to EmBERT. Note that EmBERT considers a 360-degree view, while our method only perceives a narrower 90-degree front view.

| $\mathcal{L}_{CL}$ | MA | | Seen | | Unseen | |
|---|---|---|---|---|---|---|
| | DP | Gumbel | SR | GC | SR | GC |
| | | | 34.7 | 42.0 | 3.5 | 13.6 |
| ✓ | | | 35.4 | 42.4 | 5.0 | 15.2 |
| | | ✓ | 31.5 | 38.9 | 2.7 | 12.0 |
| ✓ | ✓ | | 24.7 | 35.2 | 1.6 | 11.2 |
| ✓ | | ✓ | **36.9** | **42.8** | **8.2** | **18.0** |

Table 3: Analyses on the contrastive loss $\mathcal{L}_{CL}$ and the use of meta-actions (MA). We find that these two designs are complementary, combining them leads to the best performance. Note that during fine-tuning, the action prediction is conditioned on either the DP-labeled meta-actions or the Gumbel sampled meta-actions.

## 4.3 Ablation Studies

Following the same evaluation procedures in Sec. 4.2, we discuss each individual contribution of the contrastive loss $\mathcal{L}_{CL}$ and the use of meta-actions. We present the results in Table 3. The findings demonstrate the mutual benefit of these design choices, leading to the best performance.

**Contrastive Objective** Regarding the contrastive objective ($\mathcal{L}_{CL}$), we observe a slight improvement in model performance when using it alone, as shown in the second row of the table. The results confirm our motivation that aligning actions to instructions can enhance the agent's generalizability in the *unseen environments*. Furthermore, the results in the third row demonstrate that without $\mathcal{L}_{CL}$, model would become overly reliant on the meta-actions as they are highly correlated to the action sequences. Model may learn a degenerate solution which rely solely on the meta-action for predicting actions, ignoring other relevant information.

**Meta-Actions** In Table 3 we show that the use of meta-actions can further improved the performance with proper regularization from $\mathcal{L}_{CL}$. We hypothesize that this is because the proposed meta actions encapsulate higher-level semantics that exhibit an intuitive alignment with the instructions. The notable improvements observed in the unseen domain further validate our hypothesis that this aligning nature facilitates a better comprehension of language within our model. As a result, our model demonstrates more effective action prediction when it comes to generalizing across diverse environments. The observed performance degradation when using meta actions alone can be at-

| Method | Seen | | Unseen | |
|---|---|---|---|---|
| | SR | $\Delta(\downarrow)$ | SR | $\Delta(\downarrow)$ |
| E.T. | 34.7 | - | 3.5 | - |
| w/o instructions | 22.0 | -12.7 | 0.8 | -2.7 |
| LACMA | 36.9 | - | 8.2 | - |
| w/o instructions | 0.0 | **-36.9** | 0.0 | **-8.2** |

Table 4: Model's performance on validation split when removing sub-goal instructions from input at inference. $\Delta$ denotes the SR gap after the removal. Smaller gap indicates model being less sensitive to language instructions when predicting actions.

| | Seen | Unseen |
|---|---|---|
| E.T. | 48.2 | 47.7 |
| LACMA | 79.7 | 79.1 |

Table 5: Results from Instruction Perturbation. We assess how effectively the model alters its output in response to instruction perturbations.

tributed to a phenomenon akin to what we elaborated upon in Section 4.4. In the absence of the contrastive objective, the model tends to overly depend on visual cues to predict meta actions. Importantly, as the action prediction process is conditioned on meta actions, any inaccuracies originating from the over-dependence on visuals can propagate through the system, resulting in an undesired reduction in performance.

## 4.4 Instruction-Sensitive EIF Agents

To confirm our hypothesis that current models lack sensitivity to changes in instructions, we performed experiments where models were given only the task goal description $G$ while excluding all sub-goal instructions $S_{1:N}$. The results are presented in Table 4. In addition, we evaluate how well our models alters its output in response to instruction perturbations and report the results in Table 5. The combing results suggest that the proposed LACMA is more sensitive to language input. This reinforces our aim of aligning instructions with actions, thereby mitigating model's over-reliance on visual input and enhancing the trained agents' generalization.

## 4.5 Language Aligned State Representations

In order to assess the alignment between the learned state representations and language instructions, we conducted a probing experiment using

| Method | Seen | | Unseen | |
| --- | --- | --- | --- | --- |
| | R@1 | R@5 | R@1 | R@5 |
| Retrieve from 100 instructions | | | | |
| E.T. | 91.5 | 99.9 | 87.9 | 99.8 |
| LACMA | **96.8** | **100.0** | **91.2** | **99.9** |
| Retrieve from 1k instructions | | | | |
| E.T. | 62.3 | 97.8 | 48.0 | 94.6 |
| LACMA | **81.3** | **99.7** | **60.0** | **98.1** |
| Retrieve from 5k instructions | | | | |
| E.T. | 33.0 | 81.8 | 28.2 | 77.3 |
| LACMA | **53.9** | **97.3** | **46.4** | **94.3** |

Table 6: Results of probing the learned state representations on validation split. We probe the models using a instruction-retrieval task. R@1 and R@5 refer to Recall at 1 and Recall at 5, respectively.

a retrieval task. The purpose of this experiment was to evaluate the model's capability to retrieve the appropriate sub-goal instructions based on its state representations. To accomplish this, we followed the process outlined in Section 2.2, which extracts the state representations $z_t^v$ and pairing them with the corresponding instruction representations $z_{pos(t)}^w$. Subsequently, we trained a single fully-connected network to retrieve the paired instruction, with a training duration of 20 epochs and a batch size of 128.

During the testing phase, we progressively increased the difficulty of the retrieval tasks by varying the number of instructions to retrieve from: 100, 1,000, and 5,000. The obtained results are presented in Table 6. Notably, our method achieved superior performance across both seen and unseen environments compared to the baseline approach. Specifically, at the retrieval from 5000 instructions, our model surpasses E.T.'s recall at 1 by 20.9% and 18.2% on seen and unseen split, respectively. For the retrieval tasks involving 100 and 1,000 instructions, our method consistently outperforms E.T., demonstrating its effectiveness in aligning state representations with language instructions.

In addition, we also provide a holistic evaluation of instruction fidelity using the metrics proposed in Jain et al. (2019). Specifically, we calculate how well does the predict trajectories cover the ground-truth path, and report path coverage (PC), length scores (LS), and Coverage weighted by Length Score (CLS) in Table 7. From the provided table

| Method | Seen | | | Unseen | | |
| --- | --- | --- | --- | --- | --- | --- |
| | PC | LS | CLS | PC | LS | CLS |
| E.T. | 90.1 | 66.7 | 60 | 82.4 | 50.1 | 41.3 |
| LACMA | **92.3** | **70.8** | **65.4** | **85.4** | **52.3** | **45.3** |

Table 7: Fidelity of the generated trajectories. We evaluate path coverage (PC), length scores (LS), and coverage weighted by length score (CLS) on the ALFRED validation set.

one can see that our approach consistently outperforms E.T. across all categories. This suggests that LACMA excels in following instructions and demonstrates a stronger grasp of language nuances compared to E.T.

These results highlight the capability of our approach to accurately retrieve the associated instructions based on the learned state representations. By introducing the contrastive objective, our model demonstrates significant improvements in the retrieval task, showcasing its ability to effectively incorporate language instructions into the state representation.

### 4.6 Qualitative Results

We visualize the learned meta-actions and the retrieved instructions in Fig. 5. LACMA demonstrates high interpretability since we can use the state representation to retrieve the currently executing sub-goal. It is worth noting that while our model may produce a meta-action sequence that differs from the DP-annotated one, the generated sequence remains valid and demonstrates a higher alignment with the retrieved instruction. This behavior stems from our training approach, where the model is not directly supervised with labeled meta-actions. Instead, we optimize the meta-action predictors through a joint optimization of the contrastive objective $\mathcal{L}_{CL}$ and the action loss $\mathcal{L}_A$. Consequently, our model learns meta-actions that facilitate correct action generation while also aligning with the provided language instructions. Due to page limit, we visualize more trajectories in appendix A.6.

### 5 Conclusion

In this paper, we propose LACMA, a novel approach that addresses the semantic gap between high-level language instructions and low-level action space in Embodied Instruction Following. Our key contributions include the introduction of contrastive learn-

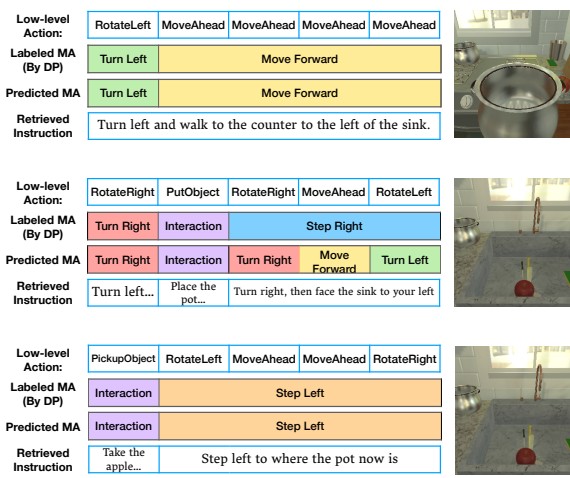

Figure 5: Visualization of the learned meta-actions and the retrieved low-level instructions. Segments with different color indicating different types of meta-actions.

ing to align the agent's hidden states with instructions and the incorporation of meta-actions, which capture higher-level semantics from the action sequence. Through these innovations, we achieve a significant 4.5% absolute gain in success rate on unseen environments. Our results demonstrate the effectiveness of LACMA in improving alignment between instructions and actions, paving the way for more robust embodied agents.

## Limitations

Despite the effectiveness of meta-actions and the contrastive objective in our approach, there are several limitations to consider. One key limitation is the use of a ResNet-50 encoder to extract a single feature for each frame. By pooling the entire image into a single vector, there is a potential loss of fine-grained information. To address this limitation, incorporating object-aware or object-centric features could enhance the model's performance. By considering the specific objects present in the environment, the model may gain a more nuanced understanding of the scene and improve its ability to generate accurate and contextually relevant actions. Another limitation is that our model does not employ any error escaping technique like backtracking (Zhang and Chai, 2021; Ke et al., 2019) or replanning (Min et al., 2022). These techniques have shown promise in improving the model's ability to recover from errors and navigate challenging environments. By incorporating an error recovery mechanism, our model could potentially enhance its performance and robustness in situations where navigation plans fail or lead to incorrect actions.

## Ethics Statement

Our research work does not raise any significant ethical concerns. In terms of dataset characteristics, we provide detailed descriptions to ensure readers understand the target speaker populations for which our technology is expected to work effectively. The claims made in our paper align with the experimental results, providing a realistic understanding of the generalization capabilities. We thoroughly evaluate the dataset's quality and describe the steps taken to ensure its reliability.

## Acknowledgements

We thank anonymous reviewers, Po-Nien Kung, Te-Lin Wu, Zi-Yi Dou and other members of UCLA-NLP+ group for their helpful comments. This work was partially supported by Amazon AWS credits, ONR grant N00014-23-1-2780, and a DARPA ANSR program FA8750-23-2-0004. The views and conclusions are those of the authors and should not reflect the official policy or position of DARPA or the U.S. Government.

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

| Meta-Actions | Regular Expressions | Example Action Sequences |
|---|---|---|
| Step Right | "rm{,3}l" | r, r, m, l, l |
| Step Left | "lm{,3}r" | l, m, m, r |
| Move Forward | "m{1,}" | m, m, m |
| Step Back | "(ll\|rr)m+(ll\|rr)" | l, l, m, r, r |
| Turn Left | "ll" | l |
| Turn Right | "rl" | r |
| Turn Around | "(lm?l)\|(rm?r)" | l, l, r, r |
| Look Up | "u{1,}" | u, u, u, |
| Look Down | "d{1,}" | d, d, d |
| Interaction | "i" | i, i, i |

Table 8: Full list of meta-actions. We use 10 meta-actions throughout our experiments.

| Low-level Actions | Letter Expression |
|---|---|
| MoveAhead | m |
| RotateRight | r |
| RotateLeft | l |
| LookUp | u |
| LookDown | d |
| PickupObject | i |
| PutObject | i |
| ToggleObjectOn | i |
| ToggleObjectOff | i |
| CloseObject | i |
| OpenObject | i |
| SliceObject | i |

Table 9: The letter expression of low-level actions, we translate the entire action sequence into the string based on the rule presented in the table.

# A  Appendix

## A.1  Implementation Details

**Model Architecture**  We use Episodic Transformer (E.T.; Pashevich et al., 2021) as our backbone. We first extract input from different modalities using modality-specific encoders, followed by a multi-modal Transformer encoder to fuse and reason over the multi-modal input. Specifically, we use a BERT-base (Devlin et al., 2019) encoder to extract features from the language instructions. For visual observations, we pre-train a ResNet-50 Faster R-CNN (Girshick, 2015) on the ALFRED dataset and use the ResNet backbone to extract image features. Note that we do not update the visual backbone during our training, instead, we use 2 convolution 1 by 1 layers, followed by a fully-connected (FC) layer, to project the features from ResNet into an embedding of the size 768. To handle actions, we train a lookup table that maps a discrete action to a 768-dimensional embedding. The multi-modal encoder comprises 2 transformer encoder layers, each with 12 self-attention heads, and a hidden size of 768, will take the aforementioned features from each modality, and produce the final state representations. We then use two separate FC layers for action and meta-action prediction. Following Pashevich et al. (2021), we use three different kinds of masking strategies for input from different modalities.

**Masking Strategy**  Specifically, the language input can only attend to ourselves, it has no access to the image and action input. The visual input can attend to all text features, but we use causal masks

to prevent them from seeing the future frames and actions. In a similar spirit, we apply the same masking strategy to the action input.

**Training parameters**  We train our model for 20 epochs in both pre-training and fine-tuning phases. The learning rate for both phases starts at 0 and linearly warms up to $1 \times 10^{-4}$ for the first 1000 steps, and drops to $1 \times 10^{-5}$ after 10 epochs. The effective batch size is 32, and we utilize 4 NVIDIA 1080Ti GPUs for training.

## A.2  Full List of Meta-Actions

We provide the full list of meta-actions in Table 8. We use letters to represent low-level actions, and we present the translate rule in Table 9. The average length of the original low-level action trajectories is around 50. While the average length of the meta-action sequences after translation is around 10, which effectively reduce the complexity of solution space. The average branching factor of low-level actions is $12^{50} \approx 10^{53}$ (50 average steps for 12 low-level actions), while for meta-action it is $10^{10}$.

## A.3  Dynamic Programming for Meta-Action Identification

Here we present the details of using dynamic-programming to identify the optimal meta-action sequences. To perform DP, we begin by determining the valid interval for each meta-action. Algorithm 1 outlines the pseudo-code for this process. We initialize a table to store the intervals associ-

---

**Algorithm 1** Finding Valid Interval For Meta-Actions

---

1:  **procedure** CREATEMETAACTIONTABLE
2:      $A \leftarrow$ low-level action sequences
3:      $M \leftarrow$ set of possible meta-actions
4:      MATable $\leftarrow$ table of size $(|M|, |A|, |A|)$ initialized with 0
5:      **for** $i \leftarrow 1$ to $|M|$ **do**
6:          interval $\leftarrow$ re.finditer$(M[i], A)$
7:          **for** $j \leftarrow 1$ to $|\text{interval}|$ **do**
8:              start, end $\leftarrow$ interval$[j]$
9:              **if** $M[i] == $ *moveahead* **then**
10:                 MATable$[i][\text{start}][\text{start} : \text{end}] \leftarrow 1$
11:             **else**
12:                 MATable$[i][\text{start}][\text{end}] \leftarrow 1$
13:             **end if**
14:         **end for**
15:     **end for**
16:     **return** MATable
17: **end procedure**

---

**Algorithm 2** Dynamic Programming for Meta-Action Identification

---

1:  **procedure** IDENTIFYMETAACTIONS
2:      $A \leftarrow$ low-level action sequences
3:      $M \leftarrow$ set of possible meta-actions
4:      $DP \leftarrow$ array of size $|A|$ initialized with $\infty$
5:      $DP[0] \leftarrow 0$
6:      MATable $\leftarrow$ CREATEMETAACTIONTABLE$(A, M)$
7:      MetaActions $\leftarrow$ array of size $|A|$ initialized with $-1$
8:      **for** $i \leftarrow 1$ to $|A|$ **do**
9:          **for** $j \leftarrow 1$ to $|M|$ **do**
10:             **for** $k \leftarrow 1$ to $|M|$ **do**
11:                 **if** MATable$[i][j][k] == 1$ **then**
12:                     **if** $DP[i] + 1 \leq DP[j + 1]$ **then**
13:                         $DP[j + 1] \leftarrow DP[i] + 1$
14:                         MetaActions$[j + 1] \leftarrow$ MetaActions$[i]$.copy()
15:                         MetaActions$[j + 1]$.append$(k)$
16:                     **end if**
17:                 **end if**
18:             **end for**
19:         **end for**
20:     **end for**
21:     **return** MetaActions$[-1][1 :]$
22: **end procedure**

---

ated with each meta-action. Using regular expressions, we identify all matching intervals for the meta-actions. If MATable$[i][j][k]$ is equal to 1, it indicates that the $i$-th meta-action is valid from the $j$-th action to the $k$-th action.

Once we have the MATable, we can perform DP to find the optimal meta-action sequences, as detailed in Algorithm 2. We initialize a dynamic programming table with the length of the transformed action sequence. Each cell in the table represents the optimal meta-action sequence up to that point. We iterate through the table, starting from the first cell, and update each cell by considering all possible meta-actions that match the corresponding sub-

| Method | Seen | | | | Unseen | | | |
|---|---|---|---|---|---|---|---|---|
| | Val | | Test | | Val | | Test | |
| | SR | GC | SR | GC | SR | GC | SR | GC |
| SEQ2SEQ (Shridhar et al., 2020) | 2.1 | 7.0 | 2.0 | 6.3 | 0.0 | 5.1 | 0.1 | 4.3 |
| MOCA (Singh et al., 2021) | 19 | 26.4 | 19.5 | 26.3 | 3.2 | 10.4 | 4.2 | 11.2 |
| EmBERT (Suglia et al., 2021) | **28.8** | **36.4** | 23.4 | 31.3 | 3.1 | 9.3 | 3.6 | 10.4 |
| E.T.[†] (Pashevich et al., 2021) | 24.6 | 31.0 | 20.1 | 27.8 | 1.8 | 8.0 | 2.6 | 8.3 |
| LACMA | 27.5 | 33.8 | **24.1** | **31.7** | **5.1** | **12.2** | **5.8** | **13.5** |

Table 10: Path-Length Weighted (PLW) results on ALFRED. Note that SR and GC denote task success rate and goal condition success rate, respectively. ([†]: trained without using data from unseen environments.)

| Method | Seen | | Unseen | |
|---|---|---|---|---|
| | SR | GC | SR | GC |
| LACMA | 36.9 | 42.8 | 8.2 | 18.0 |
| LACMA + backtrack | 37.1 | 43.8 | 10.2 | 20.6 |

Table 11: Results of applying naive backtracking technique to the proposed LACMA.

string of the transformed action sequence. Among these meta-actions, we select the one that will lead to the minimal use of meta-actions so far, and update the current cell with this optimal meta-action sequence, along with the number of meta-actions used. We gradually fill the dynamic programming table until we reach the end of the sequence. Finally, the DP algorithm traces back through the table to retrieve the optimal meta-action sequence.

## A.4 Path-Length Weighted (PLW) Scores on ALFRED

Path-length weighted (PLW) scores for vision-and-language navigation are proposed in Anderson et al. (2018a). The path-weighted score $s_p$ is defined as:

$$s_p = s \times \frac{L}{max(L, \hat{L})} \qquad (2)$$

where $L$ denotes the path length of the ground-truth trajectories, and $\hat{L}$ represents the length of the predicted paths.

From Table 10 we can observe consistent performance trends as reported in the Table 2, where our model substantially improves the task performance in the *unseen* environments in terms of success rate (SR) and goal condition success rate (GC).

## A.5 Preliminary Investigation on Backtracking

Since our proposed contrastive learning and meta actions are orthogonal to backtracking, we believe that incorporating backtracking would further improve the performance of our LACMA. From Table 11 we can see that LACMA can be extended to incorporate backtracking and further improve the success rate. Specifically, we first use E.T. to predict a sequence of subgoals and input them into LACMA. If the interaction subgoal fails, we revert to the preceding navigation subgoal. We believe further study on more sophisticated BT methods can be interesting future works.

## A.6 More Qualitative Results of the learned Meta-Actions

We provide more results in Fig. 6. The visualization further illustrates the learned meta-actions and their alignment with the retrieved instructions. Despite potential variations from the annotated meta-action sequences, the generated meta-action sequences remain valid and demonstrate a strong correspondence to the language instructions. These supplementary visualizations provide a comprehensive view of the effectiveness and robustness of our approach.

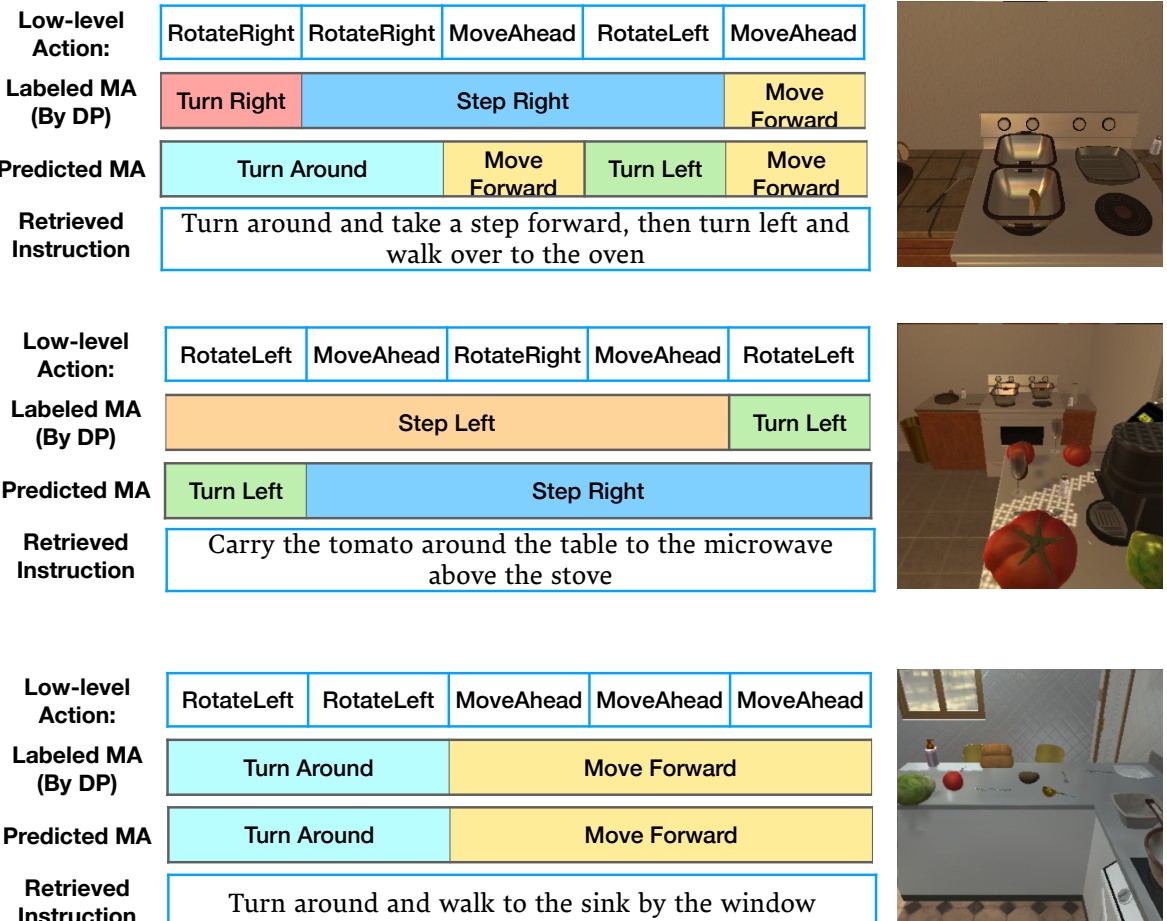

Figure 6: Visualization of the learned meta-actions and the retrieved low-level instructions. Segments with different color indicating different kinds of meta-actions.