# OpenReview forum: "LACMA: Language-Aligning Contrastive Learning with Meta-Actions for Embodied Instruction Following"
_EMNLP/2023/Conference — EMNLP 2023 Main_

### Official Review · Reviewer_3Xca · 2023-08-04

**Typos Grammar Style And Presentation Improvements:** L54
**Soundness:** 4

**Excitement:**

3: Ambivalent: It has merits (e.g., it reports state-of-the-art results, the idea is nice), but there are key weaknesses (e.g., it describes incremental work), and it can significantly benefit from another round of revision. However, I won't object to accepting it if my co-reviewers champion it.

**Missing References:**

See above.

**Paper Topic And Main Contributions:**

This paper tackles the Embodied Instruction Following (EIF) challenge. The authors propose to improve Episodic Transformer (ET), an end-to-end architecture for EIF, by introducing two components: 1. apply contrastive learning between the latent state representation used to predict action against the latent representation of the language instruction; 2. make action prediction in ET hierarchical by predicting high-level "meta actions" first: meta actions are commonly-appeared action sequence patterns summarized from training trajectories.

The first contrastive learning point is motivated by the observation that agents sometimes don't faithfully follow the language instruction but can still complete the task by memorizing patterns / taking shortcuts. This problem has been spotted before and studied before (contrary to the authors' claim it is a newly identified problem, see "Weakness 3" below), but its simplicity and effectiveness could still be of interest to the readers.

The second point on hierarchical modeling of action, however, has been extensively studied by methods on EIF in both end-to-end and modular solutions and has proven to be useful. But in this paper, the results do not seem to justify the effectiveness of the proposed "meta actions" (see more discussion below).

**Reasons To Accept:**

S1: Contrastive learning between language and action is intuitive, simple to implement, and effective. Other systems, modular or end-to-end, could benefit from having this objective in their pipelines.

S2: Clear presentation of the method. Easy to understand.

**Reasons To Reject:**

W1: The effectiveness of meta-actions (MA) cannot be justified by the experiments and analysis. Table 3 shows some conflicting messages: adding MA to Episodic Transformer (ET) decreases the performance significantly (under both DP and Gumbel settings), but when the contrastive loss is present, MA slightly increases the performance. The discussion in L470-484 does not analyze the root cause of this nor gives any hypothesis, making one doubt if MA is useful at all.

W2: There are some logical flaws in the Ablation Studies section (Sec 4.3).
- Logic gap in table 4: Authors suggest that because there is a smaller gap for LACMA (this paper's method) before and after dropping language instruction from input compared to E.T., E.T. is thus "not incorporating language instructions when predicting actions" (L489). I'm afraid I am not convinced by this. LACMA's performance drops to zero while E.T. doesn't only indicate that the lack of language destroys LACMA's latent space more than it destroys E.T.'s latent space, which makes sense, b/c LACMA has an explicit object to align the latent space to the language input, but from this, we cannot deduce that E.T. doesn't incorporate language instructions; in fact, I feel it kind of proves the opposite of the what the author claims: E.T's performance is somewhat sensitive to the language input and that's a sign of it learned some information from the language.
- Logic flaw in Table 5: better retrieval only indicates the latent space is closer, but it is a leap to claim that the action prediction would benefit from these closer-to-language latent vectors than the vectors used by E.T.

Here is an alternative ablation study proposal that I think could help prove that LACMA "understands"/"follows" language instruction: for a given language instruction, apply some perturbation (change "left" to "right", "forward" to "backward"), does the model's predicted action change accordingly? Maybe ET would be less sensitive and LACMA is more sensitive? If so I think there is a stronger evidence that LACMA can understand language better.

W3 (not a major weakness): Agent not following instructions faithfully has been studied before. This has been termed "Instruction Fidelity". See: [1] and [2]. I suggest delete the claim at L131 "We are the first ...".

W4 (not a major weakness): Contrastive learning between trajectory and language instruction has been studied before. See [3].

W5: HiTUT [4] is also an end-to-end solution with hierarchical modeling of subgoals and actions and therefore is very relevant to the topic of this paper. I'd like to see more discussions and comparisons with it.

[1] "Stay on the Path: Instruction Fidelity in Vision-and-Language Navigation" (https://arxiv.org/abs/1905.12255)

[2] "General Evaluation for Instruction Conditioned Navigation using Dynamic Time Warping" (https://arxiv.org/abs/1907.05446)

[3] "Contrastive Instruction-Trajectory Learning for Vision-Language Navigation" (https://arxiv.org/abs/2112.04138)

[4] "Hierarchical Task Learning from Language Instructions with Unified Transformers and Self-Monitoring" (https://arxiv.org/abs/2106.03427)

**Reproducibility:**

4: Could mostly reproduce the results, but there may be some variation because of sample variance or minor variations in their interpretation of the protocol or method.

**Reviewer Confidence:**

5: Positive that my evaluation is correct. I read the paper very carefully and I am very familiar with related work.

---

> ### Author Rebuttal · Authors · 2023-08-28
>
> We appreciate Reviewer 3Xca's valuable suggestions. We are motivated to see that the reviewer finds our work (1) easy to understand, (2) intuitive, sample, and effective, and (3) could benefit other navigation methods by introducing the proposed contrastive objective.
>
>
> > **Q1.** The effectiveness of meta-actions (MA) cannot be justified by the experiments and analysis. Table 3 shows some conflicting messages: adding MA to Episodic Transformer (ET) decreases the performance significantly (under both DP and Gumbel settings), but when the contrastive loss is present, MA slightly increases the performance. The discussion in L470-484 does not analyze the root cause of this nor gives any hypothesis, making one doubt if MA is useful at all.
>
> **A1.** We thank the reviewer for giving us the chance to clarify this issue.
>
> In Table 3 we show that with proper regularization (the contrastive objective), the model's performance can be further improved comparing to using the contrastive objective alone. For the sake of clarity, we present the results from Table 3 below for discussions.
>
> | Contrastive objective |  Meta Actions  |   Seen   |          | Unseen  |        |
> |:---------------------:|:--------------:|:--------:|:--------:|:-------:|:------:|
> |                       |                |  **SR**  |  **GC**  | **SR**  | **GC** |
> |           x           |       x        |   34.7   |   42.0   |   3.5   |  13.6  |
> |           v           |       x        |   35.4   |   42.4   |   5.0   |  15.2  |
> |           v           | gumbel-sampled | **36.9** | **42.8** | **8.2** | **18.0** |
>
> We hypothesize that this is because the proposed meta actions encapsulate higher-level semantics that exhibit an intuitive alignment with the instructions. The notable improvements observed in the unseen domain further validate our hypothesis that this aligning nature facilitates a better comprehension of language within our model. As a result, our model demonstrates more effective action prediction when it comes to generalizing across diverse environments.
>
> The observed performance degradation when using meta actions alone can be attributed to a phenomenon akin to what we elaborated upon in Section 4.4. In the absence of the contrastive objective, the model tends to overly depend on visual cues to predict meta actions. Importantly, as the action prediction process is conditioned on meta actions, any inaccuracies originating from the over-dependence on visuals can propagate through the system, resulting in an undesired reduction in performance.
>
>
> > **Q2.** There are some logical flaws in the Ablation Studies section (Sec 4.3). An alternative ablation study proposal could help prove that LACMA understand language better.
>
> **A2** We are grateful for the reviewer's detailed examination of the ablation studies and the thoughtful suggestions. We'll address each concern in the following:
>
> > **Q2.1** Logic gap in table 4: Authors suggest that because there is a smaller gap for LACMA (this paper's method) before and after dropping language instruction from input compared to E.T., E.T. is thus "not incorporating language instructions when predicting actions" (L489). I'm afraid I am not convinced by this.
>
> **A2.1** We apologize if our language led to a misunderstanding. We didn't mean to imply that E.T. disregards language completely. Instead, the observation we aimed to convey was that "the smaller gap in E.T.'s performance indicates potential issue that E.T. mainly relies on visual cues to predict actions rather than considering both language and vision." This will be updated in next paper revision.
>
> > **Q2.2** Logic flaw in Table 5: better retrieval only indicates the latent space is closer, but it is a leap to claim that the action prediction would benefit from these closer-to-language latent vectors than the vectors used by E.T.
>
> **A2.2** With respect, we clarify that we do not claim "action prediction would benefit from these closer-to-language latent vectors" in Sec. 4.5. Making this claim would have been a logically flaw. The intention of this experiment was to demonstrate that our approach indeed results in "closer-to-language latent vectors than the vectors used by E.T.", providing additional insights about LACMA beyond conventional metrics.
>
> If specifc line is found to suggest this wrong implication, we are happy to update the manuscript in revision.
>
> > **Q2.3** Here is an alternative ablation study proposal that I think could help prove that LACMA "understands"/"follows" language instruction: for a given language instruction, apply some perturbation (change "left" to "right", "forward" to "backward"), does the model's predicted action change accordingly? Maybe ET would be less sensitive and LACMA is more sensitive? If so I think there is a stronger evidence that LACMA can understand language better.
>
> **A2.3** Regarding the proposed ablation study, we find it to be a valuable suggestion to test the model's sensitivity to language perturbations. We present this additional study in the table below.
>
> |       | Seen | Unseen |
> | ----- |:----:|:------:|
> | E.T.  | 48.2 |  47.7  |
> | LACMA | 79.7 |  79.1  |
>
> It is evident that LACMA exhibits a notable ability to adapt its actions based on modified language instructions, which verifies our objective of building an embodied agents that can follow instructions faithfully.
>
> As for the experimental design details, we select 100 trajectories with instructions containing phrases like "Turn Left/Right/Around" from the validation split. Subsequently, we will substitute the terms "Left/Right/Around" with alternative choices. If the model's predicted actions align with the modified instructions, we will consider the prediction accurate.
>
> In addition, we also provide a hollistic evaluation of instruction fidelity using the metrics proposed in [1]. Specifically, we report path coverage (PC), length scores (LS), and Coverage weighted by Length Score (CLS) in the table below.
>
> |                       | **Seen** |          |          | **Unseen** |          |          |
> |-----------------------|----------|----------|----------|------------|----------|----------|
> |                       | **PC**   | **LS**   | **CLS**  | **PC**     | **LS**   | **CLS**  |
> | E.T.                  |     90.1 |     66.7 |       60 |       82.4 |     50.1 |     41.3 |
> | LACMA | **92.3** | **70.8** | **65.4** |   **85.4** | **52.3** | **45.3** |
>
> From the provided table one can see that our approach consistently outperforms E.T. across all categories. This suggests that LACMA excels in following instructions and demonstrates a stronger grasp of language nuances compared to E.T.
>
> > **Q3.** (not a major weakness): Agent not following instructions faithfully has been studied before. This has been termed "Instruction Fidelity". See: [1] and [2]. I suggest delete the claim at L131 "We are the first ...".
>
> **A3.** We thank the reviewer for the suggestive comment. We concur with the suggestion to remove the claim at L131. Furthermore, in response to this suggestion, we plan to incorporate experiments related to the instruction fidelity metrics proposed in [1] (see the table above in **A2.3**) within the revised version of the paper.
>
> > **Q4.** (not a major weakness): Contrastive learning between trajectory and language instruction has been studied before. See [3].
>
> **A4.** Thanks for referencing to a relevant study involving contrastive learning in the vision language navigation domain. However, we point out there are significant differences in both the approach and outcome. Regarding the approach, Liang et al. [3] contrasted data *within the same modality* (vision-to-vision, langauge-to-language), while we constrast *between instruction and state representations*. As to the outcome, Liang et al. [3] results in improved *robustness on variations* of instructions and visual scenes. In contrast, LACMA's contrastive objective enables the agent to *faithfully follow the instructions*. We will cite their paper and include this discussion in revision.
>
> > **Q5.** HiTUT [4] is also an end-to-end solution with hierarchical modeling of subgoals and actions and therefore is very relevant to the topic of this paper. I'd like to see more discussions and comparisons with it.
>
> **A5.** Thanks for your interest in seeing more comparisons. For simplicity, we did not include detailed comparisons methods that utilize backtracking / error-recovering techniques in our original submission (L588), because it might distract the readers from our focus of instruction following.
> HiTUT explicitly models subgoal to enable backtracking (BT) during navigation, and its strong performance largely came from BT (see HiTUT's Table 4).
> We believe our proposed contrastive learning and meta actions are orthogonal to backtracking, since instruction fidelity is a distinct topic from error recovering.
> To demonstrate that, we conduct additional experiments showing that LACMA can also incorporate backtracking.
> |                       |   Seen   |          |  Unseen  |          |
> | --------------------- |:--------:|:--------:|:--------:|:--------:|
> |                       |  **SR**  |  **GC**  |  **SR**  |  **GC**  |
> | LACMA                 |   36.9   |   42.8   |    8.2   |   18.0   |
> | LACMA w/ backtracking | **37.5** | **43.8** | **10.2** | **20.6** |
>
> This illustrates LACMA can be extended to incorporate BT and further improve the success rate. We will include this study in revision.
> Regarding the implementation details, we first use E.T. to predict a sequence of subgoals and input them into LACMA. If the interaction subgoal fails, we revert to the preceding navigation subgoal. We believe further study on more sophisticated BT methods can be interesting future works.

---

### Official Review · Reviewer_UNnF · 2023-08-05

**Soundness:** 4

**Excitement:**

3: Ambivalent: It has merits (e.g., it reports state-of-the-art results, the idea is nice), but there are key weaknesses (e.g., it describes incremental work), and it can significantly benefit from another round of revision. However, I won't object to accepting it if my co-reviewers champion it.

**Paper Topic And Main Contributions:**

The paper proposed a new method for instruction following in a simulated embodied environment. It leverages a contrastive learning between action and visual input history, versus the instruction embedding. It further proposed a two-stage training by predicting meta actions as well as low level actions. The meta actions were computed either through dynamic programming or Gumbel softmax. The paper experimented the methods on the ALFRED dataset and demonstrated improved performance compared to existing works.

**Questions For The Authors:**

a. The method on how to achieve meta actions could be explained further. Is there a fixed set of meta actions for the model to classify during training? What if two similar instructions share most of the same low level instructions but differ by one or two small steps? How scalable is the added meta-action learning step?

b. The same high level instruction "take a step left" could be applied in various environment. If the method forces the alignment between instructions (z^w) and visual action embeddings (z^v), wouldn't the method overfit the training visual data?

c. Table 3: why is the performance of (Gumbel only) and (contrastive learning + DP)  worse than no training objective at all (first line)? Ln 113, if DP reduces the performance, what is the purpose of this statement?

d. According the the qualitative examples, the "high level" instructions actually offers a lot more details than natural human instructions. For example, instead of saying "take a half step left and walk until you are about two feet from the fireplace", human would probably just say "get close to the fireplace". How would this method be able to generalize to unspecified levels of detailed instructions?

e. A lot of the time, a high level language instruction can be translated to a set of low level instructions, differing by minor details that are environment specific. Would be helpful to include an ablation study to demonstrate the importance of the visual input.

Questions on details:

f. Does the input camera observation history include all the way right before the action or after the action?

g. During contrastive learning, how are different timestamped & length instructions mapped to camera and action since L is not equal to T?

**Reasons To Accept:**

The paper introduced a new method for the embodied instruction following tasks hoping to alleviate the seen environment overfitting problem by adding a meta-action prediction. The paper designed two meta-action prediction methods to bridge the gap between high level natural language instructions and low lever instructions. The paper also proposed a contrastive learning scheme to strengthen the alignment between instructions and visual action embeddings. The paper conducted thorough experiments on the ALFRED dataset and demonstrated improved performance compared to baseline works.

**Reasons To Reject:**

Several questions below.

**Reproducibility:**

4: Could mostly reproduce the results, but there may be some variation because of sample variance or minor variations in their interpretation of the protocol or method.

**Reviewer Confidence:**

3: Pretty sure, but there's a chance I missed something. Although I have a good feel for this area in general, I did not carefully check the paper's details, e.g., the math, experimental design, or novelty.

---

> ### Author Rebuttal · Authors · 2023-08-28
>
> We thank Reviewer UNnF for the positive comments and suggestive remarks. We are excited to see that the reviewer finds our work (1) is a new method, (2) bridge the gap between instructions and actions, (3) strengthened alignment between instructions and and visual action embeddings, and (4) thorough experiments and improved performance. Please see our responses below for each raised issue.
>
> > **Q1.** The method on how to achieve meta actions could be explained further. Is there a fixed set of meta actions for the model to classify during training? What if two similar instructions share most of the same low level instructions but differ by one or two small steps? How scalable is the added meta-action learning step?
>
> **A1.** We are happy to clarify this issue. Yes, during the pre-training stage, we directly optimize the model using the DP-labeled meta action sequences (L282). In scenarios where two instructions exhibit similarity but vary slightly by one or two steps, we don't have to modify the set of meta actions, as we already include the atomic action (RotateLeft/RotateRight/MoveAhead) in the meta action set. The set of meta actions used during training are presented in Table 6 in Appendix A.2. Since DP will generate meta action sequences with minimal length, these atomic actions won't be used unless it can yield the shortest sequence.
>
> As a result, our proposed method can efficiently scale with a diverse range of trajectories. Referencing L254-L278, given a set of meta-actions, the designed dynamic programming algorithm can generate the optimal meta-action sequences to represent the trajectories. The detailed pseudo code and algorithmic breakdown can be found in Appendix A.3.
>
> > **Q2.** The same high level instruction "take a step left" could be applied in various environment. If the method forces the alignment between instructions (z^w) and visual action embeddings (z^v), wouldn't the method overfit the training visual data?
>
> **A2.** To clarify, the state representation $z^v_t$ is more than just a visual encoding --- it encompasses the comprehensive context of navigation by conditioning on full instructions $w_{1:L}$, observations up to the current time step $v_{1:t}$, and the past actions $a_{1:t-1}$ (L195-L197).
>
> By design, this objective mitigates the risk of overfitting within the training domain, leading to enhanced generalization across diverse environments. The outcome is evidenced by the significant 4.5% absolute gain in success rate within unseen environments, as shown in Table 2.
>
> > **Q3.** Table 3: why is the performance of (Gumbel only) and (contrastive learning + DP) worse than no training objective at all (first line)? Ln 113, if DP reduces the performance, what is the purpose of this statement?
>
> **A3.** We understand the particular concern of the degraded performance when meta action is used alone, and we are glad to further clarify this issue.
>
> We would like to first note that DP and Gumbel indicate that during fine-tuning stage, the action prediction is conditioned on the DP-labeled meta actions or the Gumbel sampled meta actions. Both methods are pre-trained using DP-labeled meta actions, as described in L279-L287. You can also find the depicted training process in Fig. 4. We now detail the reason why gumbel only model and contrastive + DP model perform worse than the baseline method.
>
> In L297-L310 of the original manuscript, we mentioned that conditioning on the DP-labeled meta actions for action prediction can lead to serious train-test mismatch, where meta actions are predicted rather than labeled. By leveraging Gumbel-Softmax, our model is able to condition on the predicted meta actions for action prediction during fine-tuning, reducing the train-test disparities, and also enable the meta action predictor to be jointly updated.
>
> The observed performance degradation when using only meta actions can be attributed to a phenomenon akin to what we elaborated upon in Section 4.4. In the absence of the contrastive objective, the model tends to overly depend on visual cues to predict meta actions. Importantly, as the action prediction process is conditioned on meta actions, any inaccuracies originating from the over-dependence on visuals can propagate through the system, resulting in an undesired reduction in performance.
>
> However, in Table 3 we show that with proper regularization (the contrastive objective), the model's performance can be further improved comparing to using the contrastive objective alone. For the sake of clarity, we present the results from Table 3 below for discussions.
>
> | Contrastive objective |  Meta Actions  |   Seen   |          | Unseen  |        |
> |:---------------------:|:--------------:|:--------:|:--------:|:-------:|:------:|
> |                       |                |  **SR**  |  **GC**  | **SR**  | **GC** |
> |           x           |       x        |   34.7   |   42.0   |   3.5   |  13.6  |
> |           v           |       x        |   35.4   |   42.4   |   5.0   |  15.2  |
> |           v           | gumbel-sampled | **36.9** | **42.8** | **8.2** | **18.0** |
>
> This is because the proposed meta actions encapsulate higher-level semantics that exhibit an intuitive alignment with the instructions. The notable improvements observed in the unseen domain further validate our hypothesis that this aligning nature facilitates a better comprehension of language within our model. As a result, our model demonstrates more effective action prediction when it comes to generalizing across diverse environments.
>
> > **Q4.** According the the qualitative examples, the "high level" instructions actually offers a lot more details than natural human instructions. For example, instead of saying "take a half step left and walk until you are about two feet from the fireplace", human would probably just say "get close to the fireplace". How would this method be able to generalize to unspecified levels of detailed instructions?
>
> **A4.** In fact, ALFRED training data comprises instruction of varying levels of specification, ranging from detailed step-by-step instructions to those with more general phrasing. Here are some example of general instructions: *"Move away from the counter"*,  *"go through the kitchen"*, which fall into the category of less specific directions. LACMA is able to generalize to unspecified levels of details due to the fact that each action trajectory is independently annotated by 3 annotators in ALFRED. The model is therefore trained to relate potentially different levels of instructions to the same visual input. This process equips the model to effectively navigate scenarios with both generic and detailed instructions.
>
> > **Q5.** A lot of the time, a high level language instruction can be translated to a set of low level instructions, differing by minor details that are environment specific. Would be helpful to include an ablation study to demonstrate the importance of the visual input.
>
> **A5.** Thanks for suggesting an additional experiment! We present the results in the table below.
>
> |                  |  Seen  |        | Unseen |        |
> | ---------------- |:------:|:------:|:------:|:------:|
> |                  | **SR** | **GC** | **SR** | **GC** |
> | LACMA w/o vision |   0.0  |  2.9   |  0.0   |   3.5  |
> | LACMA            |  36.9  |  42.8  |  8.2   |  18.0  |
>
> Note that the ALFRED environment requires the agent to predict an "interaction mask" based on the camera observation before it can interact with objects. Therefore, the above experiment only removes the visual input while agent navigates, i.e., not interacting with any object.
>
> One can see that the agent can barely success without visual input. After a closer examination, we found the only succesful instances all require only few navigation steps. The results demonstrate the importance of the visual information.
>
> > **Q6.** Does the input camera observation history include all the way right before the action or after the action?
>
> **A6.** Yes, the input camera observation history indeed includes all frames right up to the moment before the action is taken. During training, while we employ teacher forcing and input all frames at once, we also incorporate a causal attention mask. Similar to autoregressive language models, this mask ensures that the model does not have access to information from future frames.
>
> > **Q7.** During contrastive learning, how are different timestamped & length instructions mapped to camera and action since L is not equal to T?
>
> **A7.** Thanks for raising a question to one important implementation detail! During training, we establish and maintain a mapping denoted as $pos(t)$ (L209). This mapping ensures that each frame at timestamp $t$ is correctly aligned with its corresponding language instructions. Note that this alignment can be derived from the official ALFRED annotations.

---

### Official Review · Reviewer_jyH4 · 2023-08-06

**Soundness:** 3

**Excitement:**

3: Ambivalent: It has merits (e.g., it reports state-of-the-art results, the idea is nice), but there are key weaknesses (e.g., it describes incremental work), and it can significantly benefit from another round of revision. However, I won't object to accepting it if my co-reviewers champion it.

**Paper Topic And Main Contributions:**

This paper incorporates contrastive learning and meta actions (mid level action semantics) in a neuro-symbolic architecture for embodied instruction following. The main objective is to enable agents to follow instructions faithfully. The model is trained and evaluated on the ALFRED benchmark.

**Questions For The Authors:**

See above

**Reasons To Accept:**

-	Disentangling learning to *follow language instructions” from learning to “perform tasks” by following language instructions can provide a better understanding of agents’ abilities.

**Reasons To Reject:**

-	The baseline models used for comparison are much earlier models for the ALRED challenge. In the last two years, many models have been developed that have pushed the results far beyond the reported results by this work.  One of the conclusions “Compared to a strong multi-modal Transformer baseline, we achieve a significant 4.5% absolute gain in success rate at the unseen environments of ALFRED Embodied Instruction Following”. This is hardly impressive if that’s the focus (check the leaderboard).
-	In fact, the current set up in ALFRED may not serve to address the fidelity of instruction following, especially evaluation metrics. In VLN, researchers have looked at similar problems.  Jain et., 2019 cited by the paper looked at the fidelity of following language instructions. A similar set up on fidelity might be helpful here.
-	Meta actions proposed in the paper concern navigation actions. Its use might be limited to manipulation actions especially if the patterns will need to be defined before training.  Just for the navigation task, how much does it cover the ALFRED data? Any errors during inference time are attributed to insufficient coverage of patterns?
-	Any reason the path length weighted metrics are not reported in the paper?

**Reproducibility:**

4: Could mostly reproduce the results, but there may be some variation because of sample variance or minor variations in their interpretation of the protocol or method.

**Reviewer Confidence:**

4: Quite sure. I tried to check the important points carefully. It's unlikely, though conceivable, that I missed something that should affect my ratings.

---

> ### Author Rebuttal · Authors · 2023-08-28
>
> We appreciate Reviewer jyH4's valuable suggestions and their recognition of our work's value on *disentangled learning for better understanding of EIF agents*. Please see our responses below for each raised issue.
>
>
> > **Q1.** The baseline models used for comparison are much earlier models for the ALRED challenge. This is hardly impressive if that’s the focus (check the leaderboard).
>
> **A1.** We acknowledge the advancements been made in the field and discussed them in related works. However, we would like to clarify that achieving top performance on ALFRED is *not* our focus. LACMA's core objective is an embodied agent that *faithfully follows instructions* (L62). Top ALFRED leaderboard models, on the other hand, optimizes the task success rate (SR) only, by planner-based approaches that explore the environment *regardless of the instruction*.  More specifically, planner-based methods such as FILM [1] and Prompter [2] replace the natural language instruction with predefined goal templates and then conduct extensive exploration to find the goal, ignoring the instructions that usually describe how to approach the goal. Although this achieved high SR, the agents *do not follow instructions*.
>
> This limitation becomes evident when examining the maximum achievable instruction fidelity score (CLS). For instance, in the case of FILM, its CLS is capped at 0.4 due to a path-length-weight (PLW) of 0.4, in both seen and unseen data splits. In contrast, our method achieves PLW values of 0.74 and 0.62, respectively. Consequently, the resulting CLS scores are 0.65 and 0.45 for these respective subsets.
>
> Furthermore, our claim of *better instruction following* has experimental support: LACMA successfully encodes the instruction in its internal state (Sec. 4.5), which leads to more faithful instruction following trajectories. Please see the additional faithfulness evaluation in **A2**.
>
> > **Q2**. In fact, the current set up in ALFRED may not serve to address the fidelity of instruction following, especially evaluation metrics. In VLN, researchers have looked at similar problems. Jain et., 2019 cited by the paper looked at the fidelity of following language instructions. A similar set up on fidelity might be helpful here.
>
> **A2.** Thanks for suggesting the fidelity metrics! We evaluate path coverage (PC), length scores (LS), and converage weighted by length score (CLS) on the ALFRED validation set for E.T. and LACMA. The results are summarized below.
>
> |                       | **Seen** |          |          | **Unseen** |          |          |
> |-----------------------|----------|----------|----------|------------|----------|----------|
> |                       | **PC**   | **LS**   | **CLS**  | **PC**     | **LS**   | **CLS**  |
> | E.T.                  |     90.1 |     66.7 |       60 |       82.4 |     50.1 |     41.3 |
> | LACMA                 | **92.3** | **70.8** | **65.4** |   **85.4** | **52.3** | **45.3** |
>
> From the table, it's evident that our approach consistently outperforms E.T. across all fidelity metrics. This suggests that our proposed method improves the instruction following fidelity over E.T.
>
>
>
> > **Q3.** Meta actions proposed in the paper concern navigation actions. Its use might be limited to manipulation actions especially if the patterns will need to be defined before training. Just for the navigation task, how much does it cover the ALFRED data? Any errors during inference time are attributed to insufficient coverage of patterns?
>
> **A3.** We'd like to clarify that the proposed meta actions achieve *full coverage* for navigation actions since we already include the atomic action (RotateLeft/RotateRight/MoveAhead) in the meta action set. The set of meta actions used during training are presented in Table 6 in Appendix A.2. Given any trajectories in ALFRED's navigation action space, our dynamic programming algorithm can find the optimal (shortest) meta actions to represent the trajectories (L249). As a result, these atomic actions won’t be used unless it can yield the shortest sequence. We also provide the pseudo code in Appendix A.3.
>
> > **Q4.** Any reason the path length weighted metrics are not reported in the paper?
>
> **A4.** We did not include them at submission for simplicity. Here we provide the corresponding path weighted version for each metric in parentheses.
>
> | **Method** |      **Seen**       |                     |                     |                     |    **Unseen**     |                   |                   |             |
> | ---------- |:-------------------:|:-------------------:|:-------------------:|:-------------------:|:-----------------:|:-----------------:|:-----------------:|:-----------:|
> |            |       **Val**       |                     |      **Test**       |                     |      **Val**      |                   |     **Test**      |             |
> |            |       **SR**        |       **GC**        |       **SR**        |       **GC**        |      **SR**       |      **GC**       |      **SR**       |   **GC**    |
> | SEQ2SEQ    |      3.1 (2.1)      |       10 (7)        |        4 (2)        |      9.4 (6.3)      |       0 (0)       |     6.9 (5.1)     |     0.4 (0.1)     |   7 (4.3)   |
> | MOCA       |      25.9 (19)      |     34.9 (26.4)     |     22.1 (19.5)     |     28.3 (26.3)     |     5.4 (3.2)     |    16.2 (10.4)    |     5.3 (4.2)     | 14.3 (11.2) |
> | EmBERT     | **37.4** (**28.8**) | **44.6** (**36.4**) |     31.8 (23.4)     |     39.2 (31.3)     |     5.7 (3.1)     |    15.9 (9.3)     |     7.5 (3.6)     | 16.3 (10.4) |
> | E.T.       |     34.7 (24.6)     |       42 (31)       |     28.9 (20.1)     |     36.3 (27.8)     |     3.5 (1.8)     |     13.6 (8)      |     4.7 (2.6)     | 14.9 (8.3)  |
> | LACMA      |     36.9 (27.5)     |     42.8 (33.8)     | **32.4** (**24.1**) | **40.5** (**31.7**) | **8.2** (**5.1**) | **18** (**12.2**) | **9.2** (**5.8**) | **20.1** (**13.5**) |
>
> We observe consistent performance trends as reported in the original manuscript's Table 2. We will update this table in the revision.
>
> [1] FILM: Following Instructions in Language with Modular Methods. Min et al., ICLR 2022.
> [2] Prompter: Utilizing Large Language Model Prompting for a Data Efficient Embodied Instruction Following. Inoue et al., arXiv preprint 2022.

---

### Meta-Review · Area_Chair_H8ey · 2023-09-18

**Recommendation:** 4

**Metareview:**

This paper introduces a framework combining contrastive learning and high-level meta actions computed in different ways for embodied instruction following in the ALFRED benchmark. They show improved performance on the benchmark with their method and evaluate the ablated components of their system to assess the difference in performance. The authors amended and revised some flaws in the ablation studies pointed out by reviewers which has made and I urge them to consider other points raised as well, but overall this paper is a good contribution and would raise interesting discussions in the field.

---

### Decision · Program_Chairs · 2023-10-07

**Decision:**

Accept-Main

**Comment:**

This paper introduces a framework combining contrastive learning and high-level meta actions computed in different ways for embodied instruction following in the ALFRED benchmark. They show improved performance on the benchmark with their method and evaluate the ablated components of their system to assess the difference in performance. The authors amended and revised some flaws in the ablation studies pointed out by reviewers which has made and I urge them to consider other points raised as well, but overall this paper is a good contribution and would raise interesting discussions in the field.